# Acidification and Nutrient Imbalances Drive Fusarium Wilt Severity in Banana (*Musa* spp.) Grown on Tropical Latosols

**DOI:** 10.3390/jof11090611

**Published:** 2025-08-22

**Authors:** Tao Jing, Kai Li, Lixia Wang, Mamdouh A. Eissa, Bingyu Cai, Tianyan Yun, Yingdui He, Ahmed A. El Baroudy, Zheli Ding, Yongzan Wei, Yufeng Chen, Wei Wang, Dengbo Zhou, Xiaoping Zang, Jianghui Xie

**Affiliations:** 1National Key Laboratory for Tropical Crop Breeding, Institute of Tropical Bioscience and Biotechnology, Chinese Academy of Tropical Agricultural Sciences, Haikou 571101, China; jingtao@itbb.org.cn (T.J.); likai@itbb.org.cn (K.L.); wanglixia@itbb.org.cn (L.W.); mamdouh.eisa@aun.edu.eg (M.A.E.); caibingyu@itbb.org.cn (B.C.); yuntianyan@itbb.org.cn (T.Y.); heyingdui@itbb.org.cn (Y.H.); dingzheli@zju.edu.cn (Z.D.); weiyongzan@itbb.org.cn (Y.W.); chenyufeng@itbb.org.cn (Y.C.); wangwei@itbb.org.cn (W.W.); 2Department of Soils and Water, Faculty of Agriculture, Assiut University, Assiut 71526, Egypt; 3Soil and Water Department, Faculty of Agriculture, Tanta University, Tanta 31527, Egypt; drbaroudy@agr.tanta.edu.eg

**Keywords:** Fusarium wilt, *Fusarium oxysporum* f. sp. *cubense*, soil acidity, soil fertility, banana, Hainan Island, tropical agriculture, soilborne pathogens

## Abstract

Fusarium wilt, caused by *Fusarium oxysporum* f. sp. *cubense* (FOC), remains a major constraint to global banana (*Musa* spp.) production, especially in tropical regions. Although soil conditions are known to modulate disease expression, the specific physicochemical drivers of FOC prevalence under field conditions are not well understood. This study investigated the relationships between soil properties and the Fusarium wilt incidence across 47 banana farms on Hainan Island, China, a tropical region dominated by highly weathered tropical soil (latosols). The disease incidence (%PDI) and FOC abundance were quantified, alongside key soil parameters, including the pH, organic carbon, cation exchange capacity, and macro- and micronutrient availability. The soils were predominantly acidic (mean pH 4.93), with low levels of organic carbon and exchangeable calcium (Ca) and elevated levels of available phosphorus (P), potassium (K), and magnesium (Mg). The Fusarium wilt incidence ranged from 1% to 78%, with significantly higher levels observed in younger plantations (<5 years old). Statistical analyses revealed strong negative correlations between the PDI and the soil pH, exchangeable Ca and Mg, and available K. Principal component analysis further confirmed the suppressive role of the pH and base cations in the disease dynamics. Farms older than five years exhibited better soil fertility indices and lower disease pressure, suggesting a temporal improvement in soil-mediated disease suppression. These findings underscore the critical role of soil acidification and nutrient imbalances, particularly Ca, Mg, and K deficiencies, in promoting FOC pathogenicity. Enhancing soil health offers a promising and sustainable strategy for managing Fusarium wilt in tropical banana production systems.

## 1. Introduction

Banana (*Musa* spp.) is one of the most widely cultivated and consumed fruit crops globally, ranking fourth in terms of world food production after maize, wheat, and rice [1,2]. However, its productivity is increasingly threatened by *Fusarium oxysporum* f. sp. *cubense* (FOC), a soilborne fungal pathogen that causes Fusarium wilt, commonly known as Panama disease [3,4]. The disease often emerges sporadically within banana plantations and rapidly spreads from the initial infection site, potentially affecting entire plantations [4,5]. The causal pathogen, FOC, enters the plant through root wounds or natural openings, colonizes the root cortex, and progresses into the xylem vessels [6]. Within the vascular system, the fungus proliferates and produces gums and tyloses, which block water and nutrient transport, leading to leaf yellowing, wilting, pseudostem splitting, and eventually, plant death [4,6]. The pathogen produces long-lived chlamydospores that enable survival in soil for decades, making eradication extremely difficult [4,6]. Although progress has been made, the patterns of occurrence, mechanisms of transmission, and effective control strategies for Panama disease remain poorly understood. The disease has significantly impacted banana production in key regions of mainland China, including Guangdong, Guangxi, Hainan, Fujian, and Yunnan [5,6]. According to national standards, a plantation is classified as severely affected if the Panama disease incidence (PDI) exceeds 20% [7]. Immediate management actions are advised when the PDI is at or above this threshold [7,8,9]. These include planting resistant banana cultivars, applying microbial biocontrol agents such as *Trichoderma* spp. and *Bacillus subtilis*, and optimizing soil practices through organic matter amendment, balanced fertilization, and improved drainage [7,8,9]. Understanding the relationship between soil properties and the incidence of Panama disease is critical for developing effective management strategies [8].

Despite ongoing research, the interaction between soil properties and disease severity remains complex and insufficiently characterized. Most existing studies have relied on pot experiments or isolated field trials, which often yield inconsistent results and may not accurately represent field conditions [8,10,11,12]. Therefore, comprehensive field-based investigations are essential to clarify these interactions and support the development of sustainable management approaches.

Soil types exhibit considerable heterogeneity, and studies based on single-field investigations may not adequately capture the broader factors influencing the occurrence of Fusarium wilt in banana. As a result, the issue must be addressed regionally to link current soil conditions with Panama disease activity through extensive surveys. Continuous banana cultivation can deplete key nutrients (K, Ca, Mg, B), lower soil organic matter, and cause acidification, all of which may increase the disease incidence and severity [11]. The soil characteristics in Hainan have undergone significant changes in the past 30 years [13,14]. We hypothesize that suboptimal soil conditions, particularly low pH, reduced organic carbon, and imbalanced nutrient availability, are associated with increased Fusarium wilt incidence, likely due to their adverse effects on plant health and their role in enhanced pathogen activity. This study aims to investigate how soil properties, specifically the characteristics of tropical latosols, influence the incidence of FOC in banana (*Musa* spp.) plantations under tropical rainforest conditions. Additionally, we compare the fertility status of these soils against Chinese national standards to evaluate their suitability for sustainable banana cultivation. The aim is to provide a scientific foundation for improved soil management strategies and support the development of region-specific policies aimed at mitigating PDI and promoting long-term banana production sustainability.

## 2. Materials and Methods

### 2.1. Study Sites and Soil Sampling

Hainan Island is located between 108°37′–111°03′ E and 18°10′–20°10′ N, and it is the southernmost region of China. The island spans approximately 35,400 km^2^ and features a typical tropical monsoon climate, with annual rainfall ranging from 1600 to 2500 mm and an average annual temperature of 23–25 °C [13]. This study was conducted in banana plantations across four major banana-producing regions in northwest Hainan Province: Chengmai, Lingao, Baisha, and Danzhou (Figure 1). These regions are characterized by tropical latosol soils, with variations in the climate and topography, making them representative of the province’s banana-growing zones. Soil and plant sampling took place in May and June 2022, aligning with the banana fruit-hanging stage, which, based on our experience in the region and previous surveys and experiments reported in the literature, is the most suitable period for collecting samples to assess the Fusarium wilt incidence and soil conditions [3,6,7]. Within each plot, all the banana plants were assessed and categorized as either healthy or diseased based on external symptoms of Fusarium wilt (Panama disease). Typical symptoms included progressive yellowing of leaves and necrosis along the leaf margins, as illustrated in the graphical abstract. The disease assessment followed the protocols described by Shen et al. [15] and Yang et al. [16]. A total of 47 composite soil samples were collected from 47 sites across the four regions. From each site, five soil subsamples were randomly taken from the 0–30 cm soil layer within a designated 500 m^2^ area around banana plants. These subsamples were thoroughly mixed to form a single composite sample, ensuring spatial representativeness and consistency in the soil analysis.

### 2.2. Analysis of Soil Samples

Each composite soil sample was divided into two portions. One portion was kept fresh for the quantification of Fusarium pathogens, while the other was air-dried, ground, and sieved through a 2 mm mesh for physicochemical analysis. All the soil physicochemical analyses were performed according to standardized protocols outlined in the Chinese Agrochemical Soil Analysis Methods [17]. The soil organic carbon (SOC) was determined using the rapid dichromate oxidation–titration method, involving digestion with K_2_Cr_2_O_7_–H_2_SO_4_ and titration with ammonium ferrous sulfate Fe(NH_4_)_2_(SO_4_)_2_⋅6H_2_O [17]. The soil pH was determined in a 1:2.5 (*w*/*v*) soil-to-water suspension using a digital pH meter. The total nitrogen (N) content was measured by the Kjeldahl digestion method [17]. The available phosphorus (P) was extracted using 0.5 M NaHCO_3_ at pH 8.5 and quantified via the molybdenum blue colorimetric method. The exchangeable potassium (K), calcium (Ca), and magnesium (Mg) were extracted using 1 N ammonium acetate (pH 7.0). Potassium (K) and Ca were measured using flame photometry, while Mg was determined via EDTA titration. The available boron (B) was extracted using hot water and quantified using the curcumin colorimetric method. The available copper (Cu) and zinc (Zn) were extracted using the DTPA (diethylenetriaminepentaacetic acid) method and determined by atomic absorption spectrophotometry, following the procedure of Lindsay and Norvell [18]. The cation exchange capacity (CEC) was determined using the ammonium acetate saturation method. The soil fertility characteristics were evaluated against Chinese national standards [19,20], as shown in Appendix A. The evaluation aimed to assess the nutrient status and variability of banana-growing soils under a tropical monsoon climate.

### 2.3. Pathogen Quantification

At each site, the Fusarium wilt incidence was recorded as the percentage of infected banana plants showing initial symptoms (wilting and lower leaf yellowing) following Yang et al. [16]. To quantify the *Fusarium oxysporum* f. sp. *cubense* (FOC) in the soil samples, we employed a modified version of the method developed by Zhou et al. [21]. Rhizosphere soil suspensions were serially diluted and plated on Petri dishes containing a modified Komada’s selective medium. The basal medium included 10 g D-galactose, 2 g L-asparagine, 16 g agar, 0.5 g MgSO_4_, 0.5 g KCl, 1 g K_2_HPO_4_, and 0.01 g Fe-Na-EDTA dissolved in 900 mL distilled water. It was supplemented with 100 mL of a solution containing 0.9 g pentachloronitrobenzene (PCNB, 75% WP), 0.5 g sodium tetraborate (Na_2_B_4_O_7_), 0.45 g oxgall, and 0.3 g streptomycin sulfate, with the pH adjusted to 3.8 ± 0.2 using 10% (*v*/*v*) phosphoric acid. The plates were incubated at 25 °C for 10 days. Colonies with typical FOC morphology, small (0.5–1 mm), pure white, dense aerial hyphae at the center, sparse radial hyphae at the edges (1–1.5 mm), with serrated margins (Appendix A), were selected. Genomic DNA was extracted from representative colonies, and PCR verification was performed using the FOC-specific primer SIX9_FOC_F/R [22]. Mycelial DNA of a verified FOC isolate was used as a positive control. Colony counts were performed in triplicate for each composite soil sample to ensure reproducibility and accuracy.

### 2.4. Data Analysis

Statistical analyses were performed using SPSS version 17.0 (Chicago, IL, USA) and Microsoft Excel. Descriptive statistics, including mean values and standard deviations, were calculated for each soil physicochemical parameter. Pearson’s correlation analysis was performed to assess the relationships between soil properties and the percentage disease incidence (PDI) of Fusarium wilt. To explore the multivariate relationships among the soil variables and disease incidence, principal component analysis (PCA) was conducted using GraphPad Prism (version 9.5). To further assess these relationships, a Mantel test was performed using the online OmicShare tools platform (https://www.omicshare.com/tools accessed on 1 September 2023). This test evaluated the correlation between the percentage disease incidence (PDI) and soil characteristics. In the resulting network analysis, the width and color of the edges represented the Mantel’s *R* and *p*-values, respectively. Pairwise Spearman’s correlation coefficients among the soil variables were also calculated and visualized using a heatmap with a color gradient scale, illustrating the strength and direction of the inter-variable relationships.

## 3. Results

### 3.1. Properties of Tropical Banana Soil on Hainan Island

#### 3.1.1. Soil pH, Cation Exchange Capacity (CEC), and Organic C

The soil pH across the surveyed banana plantations in Hainan ranged from 3.64 to 6.79, with an average of 4.93 (Appendix A). The majority of soils (57%) had a pH of between 4.5 and 5.5, while 28% had a pH of less than 4.5 (Figure 2 and Appendix A). The soil CEC in the studied Hainan area ranged from 2.81 to 15.95, with an average of 9.78 cmol kg^−1^. The low CEC values were found in 47% of the studied area, while 53% of the area had medium CEC (Figure 2 and Appendix A). The soil organic matter ranged from 7.85 to 46.06 g kg^−1^, with an average of 23.97 (Appendix A). Only 25% of the soils investigated on Hainan Island had a high amount of soil organic matter (SOM), while 75% of Hainan soils had SOM of less than 30 g kg^−1^ (Figure 2 and Appendix A).

#### 3.1.2. Macronutrient Status

The total soil nitrogen (N) content varied from 0.33 to 1.75, with an average of 1.15 g kg^−1^ (Appendix A). Only 17% of the soil samples had a high amount of total N, compared to 83% of the samples with total N of less than 1.5 g kg^−1^ (Figure 2 and Appendix A). Only 21% of the soil had high Ca content (Figure 2 and Appendix A), whereas 47% had low Ca content (less than 500 mg kg^−1^). The P availability was high in most of the study area (89%), whereas the K availability was high in 85%. Most of the examined soils (62%) had Mg levels above the threshold. The values of the soil exchangeable Mg, available P, and K (Figure 2 and Appendix A) in the Hainan soil were higher than the Chinese standards according to the 2nd National Soil Survey.

#### 3.1.3. Micronutrient Status

Most of the studied area (62%) contained very high available soil zinc (Zn), whereas 36% of the studied area had a high Zn level (Figure 2 and Appendix A). Approximately 98% of the study area contained a sufficient amount of available soil Zn. In most of the examined soil samples, the boron (B) and copper (Cu) concentrations were medium or high. In 41% of the soil samples, there was a very high level of B, and in 40% of the soil samples, there was a high level (Figure 2). Very high levels of Cu were found in 34% of the soil samples, whereas 45% of the soil samples contained high levels.

### 3.2. Effect of Cultivation Time on Fusarium Wilt in Banana

The data in Figure 2 and Appendix A show that the infection rate of Fusarium wilt in the banana orchards in Hainan was 1–78%. According to this study, 36% of the Hainan banana plantations had infection rates under 5%, whereas 28% of the region had rates between 5 and 15%. The PDI significantly correlated with *Fusarium oxysporum* (Foc) in the soil (Figure 3). The correlation analysis (Figure 3) revealed that the FOC population in soil was strongly and positively associated with the PDI (r = 0.8802, *p* < 0.0001). Among the soil chemical properties, significant negative correlations were observed with the soil pH, exchangeable Mg, and exchangeable Ca, suggesting that acidic conditions with low base cation availability favor higher FOC populations and greater disease severity. In contrast, the available K, P, Zn, Cu, B, SOM, N, and CEC exhibited weak or non-significant correlations with FOC, indicating that these factors were less influential in determining the disease dynamics in the surveyed farms. These results highlight soil acidity and deficiencies in Ca and Mg as the main chemical drivers of Fusarium wilt outbreaks in Hainan banana plantations.

The violin charts (Figure 4 and Figure 5) indicate that farms less than five years old exhibited higher FOC abundance and greater PDI values than older farms. Moreover, the soil properties—defined here by the optimum soil pH, greater organic carbon content, higher levels of key nutrients (N, P, K, B, Zn, Ca, Mg), and greater cation exchange capacity (CEC)—were generally better in farms older than five years compared with newly established farms (1 year old), with intermediate-age farms showing variable values.

Figure 5 shows the distribution of FOC populations in the soils of banana orchards with different plantation ages and disease severities. In orchards with a PDI < 20%, the FOC populations remained consistently low across all the age groups (1, 2–5, and >5 years), with no significant differences observed. In contrast, in orchards with a PDI > 20%, the FOC populations were markedly higher, particularly in 1-year-old plantations, which showed significantly greater populations compared to the 2–5-year-old plantations. The pie charts illustrate the relative proportion of orchards in each age group within the respective disease severity categories, showing that younger orchards tended to have a higher share of severe disease incidence. These results suggest that both plantation age and disease severity strongly influence soil FOC populations, with younger farms under severe infection conditions harboring the highest inoculum loads.

### 3.3. Relationships Between Soil Properties and Fusarium Wilt in Banana

Data from the present study showed that the disease incidence was significantly and negatively correlated with the soil pH, exchangeable Ca and Mg, and available K (Figure 3). PCA (Figure 6) identified these same variables as the most influential factors associated with Panama disease severity. The exchangeable Ca and Mg also showed negative correlations with the FOC activity and PDI, suggesting that higher levels of these cations may help suppress disease. Among the essential nutrients assessed, only K, Ca, and Mg exhibited highly significant negative associations with the disease incidence (Figure 7), underscoring their importance in the soil fertility status of Hainan banana farms.

## 4. Discussion

### 4.1. Relationships Between Soil Chemical Properties and Fusarium Wilt in Banana

Our findings indicate a negative correlation between the soil pH and the Panama disease severity, suggesting that soils with optimum pH levels may have lower disease risk. A negative correlation between the soil pH and Panama disease was also reported by Fan and Li [23], Shen et al. [24], and Gatch and Toit [25]. Based on the correlation analysis between the number of FOC, PDI, and soil, the PDI increased with an increasing number of FOC, which increased under acidic soil conditions. These shifts in the pH of acidic soils are likely due to long-term soil management practices—such as liming, balanced fertilization, and organic matter amendments—which gradually improve the nutrient availability and cation exchange capacity, thereby contributing to the suppression of Panama disease [8,11]. The preferred soil pH range for bananas is 5.8 to 6.5, measured in 1:2.5 soil to water [26,27]. Although the pH range of 3.50 to 7.00 has been reported as favorable for the growth of plant-pathogenic *Fusarium* isolates, raising the pH of Hainan soils can improve nutrient availability and enhance banana root growth, thereby reducing the severity of Panama disease [10,28,29]. The extreme acidity of the soil may adversely affect the soil nutrient availability and plant growth, ultimately leading to insufficient plant resistance, given that the fungus can grow in a wide range of soil acidities [10,28,29].

Essential nutrients play a critical role in mitigating the severity of Fusarium wilt in bananas by enhancing the plant’s overall disease tolerance [30,31,32]. Previous studies have shown that altering the soil N, P, K, Mg, Mn, and Zn levels may reduce the prevalence of Fusarium disease through direct toxic effects or by influencing the cell wall composition, lignin production, phenol biosynthesis, photosynthesis, and other defense-related processes [30,33]. Nutrient deficiencies, conversely, can increase plant susceptibility to Fusarium infection; in nutrient-deprived plants, leaf tissues tend to accumulate higher levels of soluble sugars and amino acids, which serve as favorable substrates for pathogen proliferation [8,30,32]. In the present study, we also observed that higher levels of exchangeable Ca and Mg and available K were negatively correlated with both the PDI and FOC abundance, supporting the role of balanced nutrient availability in reducing disease severity in Hainan banana farms.

Hainan is experiencing substantial soil degradation, including soil erosion, soil acidity, a drop in SOC, and loss of biodiversity, in addition to the conversion of woodlands into agricultural land and switching to intensive agricultural techniques [14,34]. Hainan has one of the highest rates of water erosion in China (and worldwide), together with a high frequency of heavy rainstorms [35,36]. Inappropriate agricultural management techniques, combined with rapid soil erosion caused by heavy rainfall, have resulted in a significant loss of surface soil organic matter and vital plant nutrients [37,38].

### 4.2. Properties of Banana Tropical Soil on Hainan Island

Over the past three decades, the soil properties of Hainan have undergone substantial changes due to climatic influences, land use shifts, and agricultural intensification [13,14]. Understanding how these changes influence banana production systems and the incidence of Panama disease is critical for sustainable cultivation in the region. The current study revealed that a majority of soils in the surveyed banana-growing regions exhibited acidic to strongly acidic conditions, with 57% of the soils having a pH of between 4.5 and 5.5, and 28% falling below pH 4.5. Although bananas can tolerate a wide range of soils, they generally thrive in well-drained soil and a pH range of 5.0–7.5 [26]. Optimal banana growth is typically achieved in soils with a pH of between 5.8 and 6.5 [27]. Hainan’s soils are predominantly lateritic, covering about 53% of the island’s land area, with 21% classified as acidic and 78% as extremely acidic (pH < 4.5) [13,14,39]. The data from the current study showed that the soils on Hainan Island (85%) had a low pH (< 5.5), and similar results were reported by Hui et al. [13], Li et al. [14], and Stocking [39]. Laterite soils, typical in wet and hot tropical regions, undergo intense leaching of base cations, leading to the dominance of aluminum (Al) and iron (Fe) oxides [40]. This mineral composition contributes to the strong acidity and nutrient imbalances observed in Hainan’s banana-growing soils.

According to the 2nd Chinese National Soil Survey, the studied soils contained low levels of exchangeable calcium (Ca) [19,20]. A low soil pH (<5) is often associated with elevated aluminum (Al) and iron (Fe) oxides and reduced base cations such as Ca^2+^ [41,42,43,44]. In contrast, the magnesium (Mg), phosphorus (P), and potassium (K) levels in Hainan’s soils were generally higher than Chinese national standards [19]. Micronutrient analysis indicated that boron (B) was sufficient in approximately 96% of the study area, and copper (Cu) was sufficient in nearly all the samples [19,20]. Other micronutrients, including zinc (Zn), Cu, and B, were present at levels adequate for optimal plant growth in most sites [45,46,47,48]. The enrichment of micronutrients in these soils is likely driven by both natural factors (parent material, topography, and climate) and anthropogenic inputs, such as long-term fertilizer application [30,31].

Hainan’s soils are generally characterized by low soil organic matter (SOM), with more than 89% of the area containing less than 30 g kg^−1^ [49]. This finding is consistent with the 2nd Chinese National Soil Survey, which also reported low SOM levels in the studied soils [19,20]. Consequently, the nitrogen (N) and organic carbon contents are low, leading to reduced soil fertility, particularly in the lateritic soils of Hainan [14,50,51].

Among the key constraints to banana cultivation in Hainan’s laterite soils are the pronounced soil sodicity and inadequate SOM [19]. The high heat and humidity on Hainan Island, along with the annual soil temperature exceeding 25.5 °C, are very beneficial for soil weathering and biogeochemical cycles of elements [14,52]. These conditions promote the intense decomposition of primary silicate minerals, leading to the formation of secondary minerals such as kaolinite, along with the accumulation of Al and Fe oxides [14,50,52]. These secondary minerals shape the lateritic zonal soil, which is highly weathered, has a low cation exchange capacity, and strong acidity [14,50,52].

### 4.3. Effect of Cultivation Time on Fusarium Wilt in Banana

Interactions among banana crops, pathogens, and soil environments are significantly influenced by a number of physical, chemical, microbiological, meteorological, and sociocultural factors [8,11,47,53]. Fusarium wilt in bananas may develop because of these intricate and site-specific interactions, which negatively affect the banana yield [12,54]. It is crucial to maintain a plant disease incidence < 20% to ensure farm health [7]. The activity of FOC and the PDI infections were higher in the farms aged less than 5 years. Moreover, the soil characteristics were much higher in old farms (>5 years) than in new farms (<5 years). This observation can be explained by the consistently low levels of soil FOC and excellent soil property indices of the older farms, whereas the newer farms showed significant enrichment of FOC in the soil, as indicated by a PDI > 20%. China’s critical PDI guideline is 20%, and fields exceeding this threshold are considered severely infected [7]. In our study, the infection rate was above the critical limit in half of the newly established farms that were under a year old, and it exceeded the critical level in 67% of the farms that were between two and five years old. All farms aged more than five years had a PDI below China’s critical standard. Karangwa et al. [55] reported higher disease incidence in aged banana farms compared with younger ones. Other studies have shown that continuous banana cultivation can lead to distinct shifts in soil microbial communities. In Sub-Saharan Africa, different long-term cropping regimes led to changes in both bacterial and fungal communities, including shifts in dominant phyla such as *Proteobacteria, Actinobacteria, Acidobacteria*, and *Bacteroidetes*, with the community structure influenced by disease pressure [56]. Similarly, extended monoculture spans were associated with significant alterations in the soil microbiome composition, where higher fungal richness correlated with greater Fusarium wilt incidence [57]. Such shifts suggest that older banana plantations may develop unique microbial assemblages that influence disease dynamics [58]. In our study, older banana farms had suppressed the initial critical phase with the transmission of Fusarium wilt during the early stages of cultivation. The difference between our study and that of Karangwa et al. [40] is that the scale of age in our study was only five years, and most of the infected farms were recently initiated. Some farmers with plantations older than five years had implemented management practices to combat PDI infection, including planting resistant cultivars and applying microbial biocontrol agents such as *Trichoderma* spp. and *Bacillus siamensis* [59,60]. These included planting resistant cultivars to reduce host susceptibility and applying microbial agents to suppress FOC populations in the soil. In the current study, most of the new farms (less than five years old) were infected with the fungus. Many of these farms received little or no management after establishment, which may have contributed to the increased infection rate. Farms older than five years may have reached a stage of stable production and economic return, encouraging farmers to invest more in management practices such as sanitation, use of resistant cultivars, and application of microbial agents. These measures likely helped reduce the infection rates. In contrast, new infections in previously unaffected areas are often introduced through the movement of contaminated planting materials [43,44].

## 5. Conclusions

This study offers valuable insights into the relationship between soil chemical properties and the incidence of Fusarium wilt (Panama disease) in banana plantations across the tropical latosols of Hainan Island, China. The findings demonstrate a strong association between highly acidic soils, nutrient imbalances—particularly calcium (Ca), magnesium (Mg), and potassium (K)—and increased prevalence of *Fusarium oxysporum* f. sp. *cubense* (FOC). Notably, younger banana farms (<5 years) exhibited significantly higher levels of FOC activity, whereas older farms (>5 years) showed lower infection rates. The improved soil fertility status of older farms may be attributed to the cumulative effects of sustained land management, including regular organic matter inputs from crop residues, optimized fertilization practices over time, reduced soil disturbance, and the gradual buildup of beneficial microbial communities. These practices can enhance nutrient cycling, stabilize the soil pH, and improve microbial resilience, all of which contribute to healthier soils. In our study, older farms (>5 years) showed higher levels of exchangeable Ca and Mg, available K, and organic carbon, along with more favorable pH values, compared to newer farms. This suggests that the farm age and continuous cultivation play a critical role in improving soil health and suppressing disease outbreaks. Encouraging long-term cultivation practices, rather than frequent land turnover, may therefore be an effective strategy to reduce the Fusarium wilt incidence. By emphasizing soil chemical properties—particularly in established banana plantations—this research provides a scientific foundation for integrated disease management strategies.

## Figures and Tables

**Figure 1 jof-11-00611-f001:**
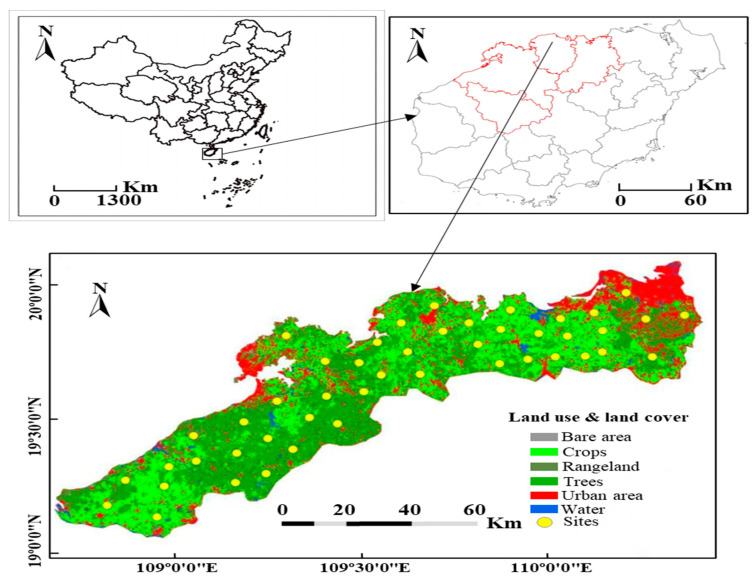
Field locations of banana orchards in the study area in Hainan.

**Figure 2 jof-11-00611-f002:**
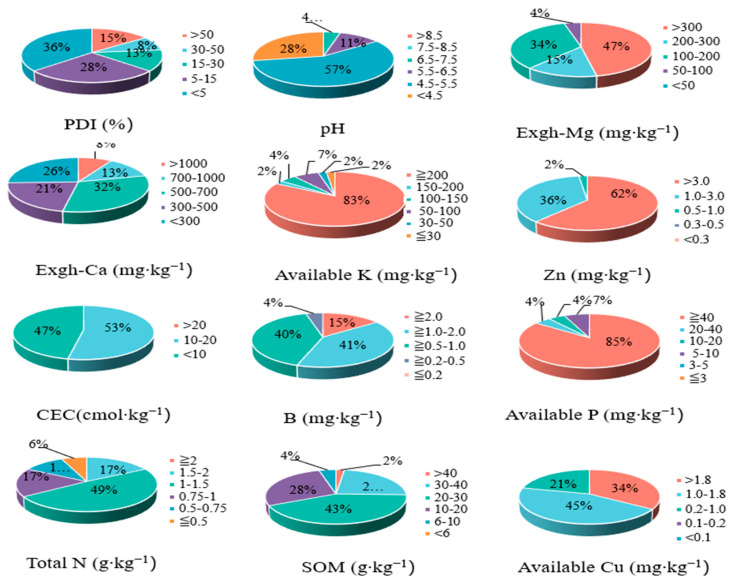
Grading of soil property parameters and Panama disease incidence (PDI) in the study area of Hainan banana orchards (*n* = 47). In each subfigure, colors indicate categories (low, medium, high, or very high) for that specific soil parameter according to Chinese national standards. Note that the color categories apply independently to each subfigure. CEC = cation exchange capacity; Exgh = exchangeable; SOM = soil organic matter.

**Figure 3 jof-11-00611-f003:**
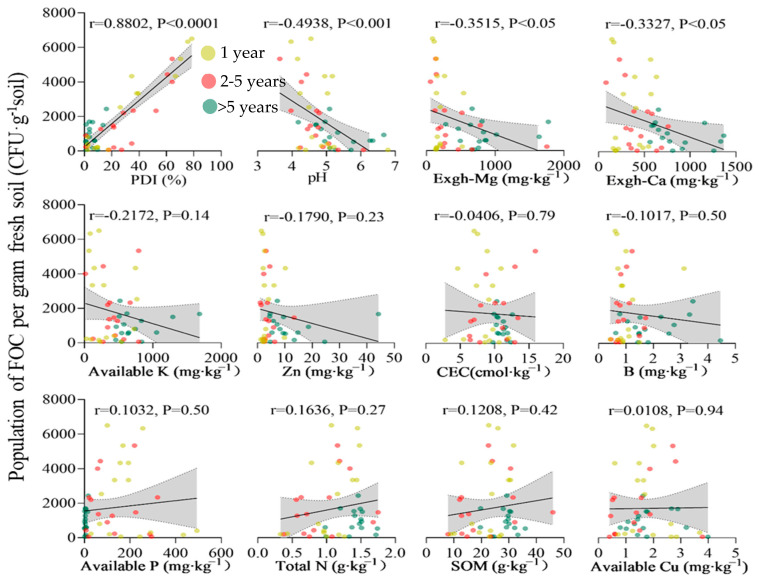
Correlations between the number of FOC and the PDI and soil chemical indexes. R represents the strength of the correlation, *p*-value represents the significance according to Pearson’s correlation test.

**Figure 4 jof-11-00611-f004:**
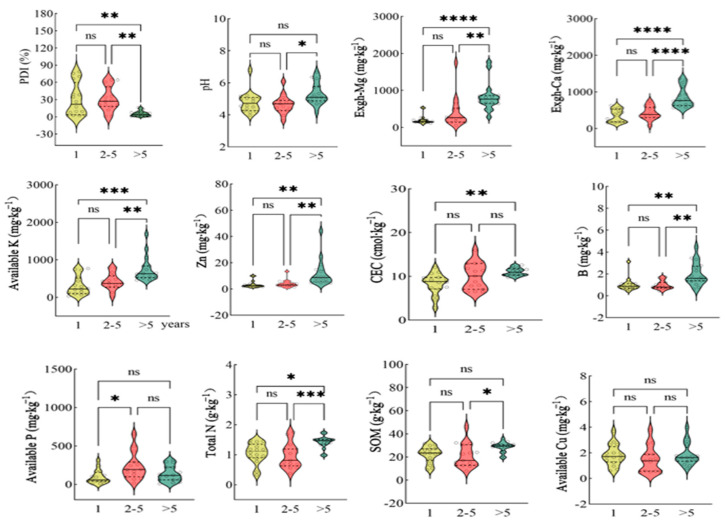
The violin chart describes the distribution of chemical indexes in the soil and the PDI of banana orchards in different planting years. The number of * indicates a significant difference at a 0.05, 0.01, 0.001, or 0.0001 significance level, and “ns” indicates no significant difference between groups according to Tukey’s post hoc tests.

**Figure 5 jof-11-00611-f005:**
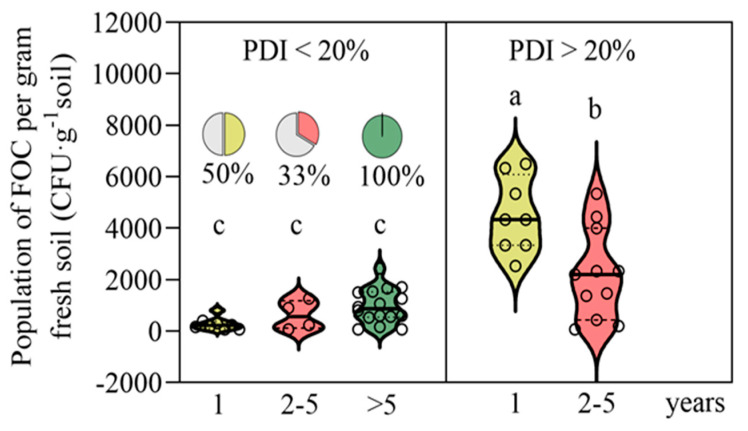
Violin plots showing the differences in soil FOC populations (CFU g^−1^ soil) among banana orchards of different ages with moderate (PDI < 20%) and severe (PDI > 20%) disease incidence. Pie charts represent the proportion of orchards in each age group relative to the total number of orchards within the corresponding disease category. Different letters denote significant differences at *p* < 0.05 according to Duncan’s multiple-range test.

**Figure 6 jof-11-00611-f006:**
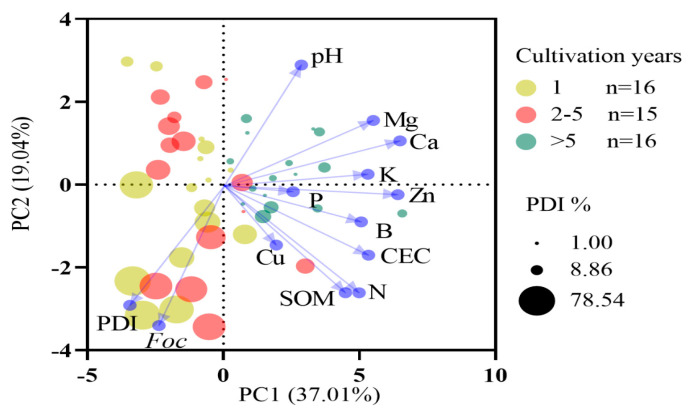
Principal component analysis (PCA) of soil fertility indicators and Panama disease index (PDI). CEC = cation exchange capacity, Exgh = exchangeable, and SOM = soil organic matter.

**Figure 7 jof-11-00611-f007:**
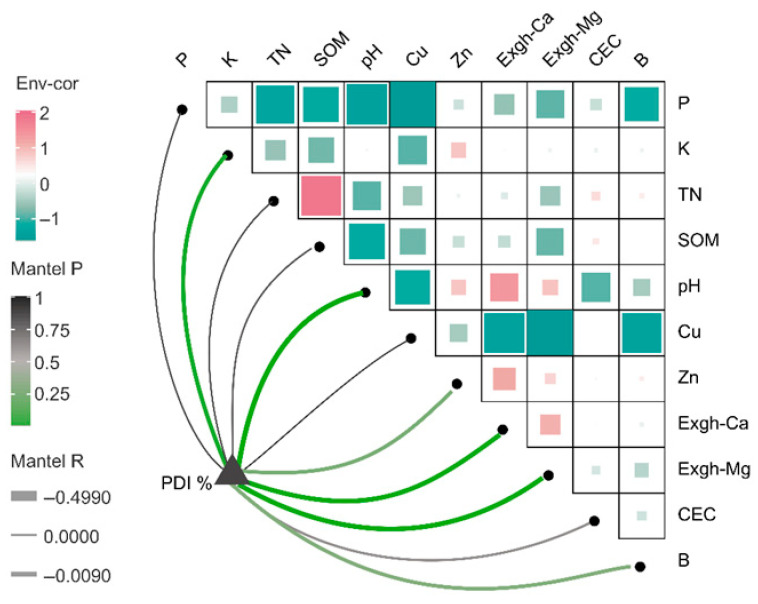
Correlations between Panama disease incidence (PDI) and soil variables based on the Bray–Curtis dissimilarity. “Env-cor” represents environmental correlations between soil variables and the PDI, shown as a color gradient where red indicates positive correlations and blue/green indicates negative correlations. Edge width corresponds to the Mantel’s *R* statistic (strength of correlation), and edge color indicates Mantel’s *p*-value (significance level). Pairwise Pearson’s correlations among the soil variables are represented in the upper matrix, with the color intensity reflecting the correlation strength and direction. CEC = cation exchange capacity; Exgh = exchangeable; SOM = soil organic matter; TN = total nitrogen.

## Data Availability

All the data used are included in the article.

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
