# Peer review of "Acidification and Nutrient Imbalances Drive Fusarium Wilt Severity in Banana (Musa spp.) Grown on Tropical Latosols"

_jof, 2025, doi:10.3390/jof11090611_

Round 1
Reviewer 1 Report
General Comments
The manuscript represents and interesting study where 47 banana farms of varying age were surveyed across Hainan, China for soil properties and incidence of Fusarium wilt TR4. The authors suggest that incidence of Fusarium wilt TR4 determined by the percentage of disease incidence (PDI) on the banana farms was related to soil pH and the exchange Ca and Mg in the soil. They make this conclusion that older banana farms, those more than 5-years old, tend to have less acid soil (closer to neutral pH 7) and greater availability Ca and Mg with lower PDI, based on correlations and Principal Component Analysis.
The conclusion by the authors would suggest that as banana plantations age they tend to have less disease due to improvements in soil properties. However, this would appear to be due to the way the authors have analysed their data, as crop age becomes a confounding factor in disease incidence. As the survey was taken at a single point in time, those banana plantations that are greater than 5-years must have had less disease in the first year of establishment to continue production. Whereas the younger plantations are unlikely to progress beyond 1-2 years due to the high disease incidence. As this study was not designed as a longitudinal study there is no way of knowing what the soil properties were like at the time of crop establishment on the now 5-year old banana fields. Likewise, there is no way of knowing what the soil properties will be like in 5-years time on the newly established plantation. It is highly unlikely that PDI would reduce the longer the banana fields were in production when they start from a point of high disease incidence. This is the conclusion that the authors appear to be suggesting? There is a need to analyse soil property data within the different age groups, that is 1, 2-5 and >5 years to determine if those with less disease in each age group have the same soil properties. It is likely that banana plantations that have been established where care has been taken to improve soil quality, neutral soil pH, adequate Ca and Mg and other nutrients are more likely to have less disease after establishment and therefore greater productive life (>5-years) than crops where soil quality has not been adjusted.
Specific comments
Abstract
L26: Emdash should be replaced with a comma
L32-33: The design of the experimental survey would means crop age has become a confounding factor or a covariate in needs to be accounted for in the analysis.
L37-39: Replace emdashes with commas.
L39-41: Amendments were not included within the study. This sentence would appear to be typical of an AI generated statement. Suggest removing this sentence.
Introduction
The introduction seems repetitive and requires reviewing to remove repetition to improve the structure and flow of information within this section.
L47: “cubense” should not have a capital.
L72-73: Replace emdashes with commas.
L75: Suggest… “This study investigated how soil properties……”
L77: This line with the full genus species name of Foc and banana appears repetitive and can use abbreviations.
L80: Suggest “The aim is to provide a….”
Material and methods
The materials and methods section appears to have repetition on how sampling was performed on L96-100 and L131-135. I would suggest that the materials and methods section requires reviewing and restructuring to remove repetition and provide a clearer description about how the work was undertaken.
L93: What is meant by the statement “diverse agroecological conditions”? The conditions need to be specified.
L95: Foc description has been repeated.
L98: Suggest using “external symptoms”.
L101-105: The soil collection description in these sentences is confusing and requires greater clarification. How much soil was collected from each site?
L109: Suggest “Each composite soil sample was divided into….”
L113: Full stop seems to be missing.
L131: Banana wilt or Fusarium wilt. This needs to be kept consistent throughout the text.
L131-135: These sentences are a repeat.
L136-146: The description of Komada’s media is selective for Fusarium oxysporum’s (Fo) and not selective for Foc. Therefore, the description and results need to be altered to indicate that only Fo was quantified and not Foc.
Results
Within the results the authors should consider investigating within crop age soil properties. As crop age appears to a confounding factor within the study, do the same soil properties that is pH, Ca and Mg, appear to be important for lower disease incidence, in 1-year old, 2-5-year old and >5-year old bananas?
L164: Suggest “Properties” rather than “Quality”
Figure 2. The colours of the different segments of the pie chart appear to be repeated on some of the charts making, if difficult determine what segments correspond to the categories in the legend. Furthermore, the order of the 12 different properties is confusing with the less important properties appearing first. I would suggest using the same order and sequence of soil properties as used in Fig. 3, as this appears to be a logical sequence. Importantly, keep the order consistent between the different figures, 2, 3 and 4.
L181: Remove the “’s” from macronutrient.
Figure 3:
- There is no description of what the different colours represent on the 12 charts. Are these different ages of plantations?
- Why use population of Fo (CFU/g soil) on the Y axis? This would be better as PDI (%) as this is the measure that was evaluated in the field. The population of Fo only indicates that Fusarium oxysporums were correlated in the disease symptoms.
- Many of the relationships do not appear to be linear, but rather an exponential relationship. An exception is the relationship between Fo populations and PDI. A linear representation maybe misleading for many soil properties. For example, above pH 6.2 there would appear to be no Fo or disease using a linear regression model.
- Therefore, the authors need to consider using PDI as the Y variable, using more appropriate relationship models and explaining the different colours on the charts.
Figure 4: The order of the charts should be kept consistent with Figure 3.
Figure 5: This chart is hard to interpret. What are the pie charts representing? The description in the caption is unclear. Is the same analysis being done across both sections of the charts? Is there a better way to represent the information.
L223: “Properties” rather than “quality”.
L225: Fig. 2 or Fig 3. This is confusing.
Figure 7: Is this a repeat of the data from Fig 3. Do these correlations assume a linear relationship, when data from Fig. 3 would suggest non-linear relationships.
Discussion
The order of the sections in the discussion should be reviewed. The more important finding should be discussed first. That is, the relationship between soil properties and disease incidence followed by general discussion about the different soil properties.
L293-314: These sentences seem to be contradictory to what happens in reality. They suggest that the older the plantation the healthier it becomes. Do young plantations that have a high incidence of Fusarium wilt become healthier if they keep growing bananas if they increase soil pH and Ca and Mg. It is more likely that those plantations that have come to be greater than 5-years old had favourable soil properties, more neutral soil pH, higher Ca and Mg, and therefore were able to maintain a low incidence of disease. However, as this was not a longitudinal study, there is no way to validate this information.
L323-331: The discussion on soil pH should be include impacts of soil pH on microbial diversity and functions. What is the likelihood that pH induced changes in microbial diversity, such as acidification have led to reduced plantation life due to increased disease?
Conclusion
L356-358: Is the longevity of banana farms due to improvements in soil properties as the plantation ages or due to improved soil properties when plantations were established. Care in how the findings are interpret due to the confounding effects of plantation age are required in this study.
L361-363: Is this emphasising soil quality in established plantations, or during plantation establishment? Replace emdashes with commas.
L363-364: How do policymakers use this information? I suggest removing this sentence.
Author Response
Reviewer-1
A-General Comments
The manuscript represents and interesting study where 47 banana farms of varying age were surveyed across Hainan, China for soil properties and incidence of Fusarium wilt TR4. The authors suggest that incidence of Fusarium wilt TR4 determined by the percentage of disease incidence (PDI) on the banana farms was related to soil pH and the exchange Ca and Mg in the soil. They make this conclusion that older banana farms, those more than 5-years old, tend to have less acid soil (closer to neutral pH 7) and greater availability Ca and Mg with lower PDI, based on correlations and Principal Component Analysis.
The conclusion by the authors would suggest that as banana plantations age they tend to have less disease due to improvements in soil properties. However, this would appear to be due to the way the authors have analysed their data, as crop age becomes a confounding factor in disease incidence. As the survey was taken at a single point in time, those banana plantations that are greater than 5-years must have had less disease in the first year of establishment to continue production. Whereas the younger plantations are unlikely to progress beyond 1-2 years due to the high disease incidence. As this study was not designed as a longitudinal study there is no way of knowing what the soil properties were like at the time of crop establishment on the now 5-year old banana fields. Likewise, there is no way of knowing what the soil properties will be like in 5-years time on the newly established plantation. It is highly unlikely that PDI would reduce the longer the banana fields were in production when they start from a point of high disease incidence. This is the conclusion that the authors appear to be suggesting? There is a need to analyse soil property data within the different age groups, that is 1, 2-5 and >5 years to determine if those with less disease in each age group have the same soil properties. It is likely that banana plantations that have been established where care has been taken to improve soil quality, neutral soil pH, adequate Ca and Mg and other nutrients are more likely to have less disease after establishment and therefore greater productive life (>5-years) than crops where soil quality has not been adjusted.
Author response:
We sincerely thank the reviewer for this detailed and thoughtful assessment of our study design and interpretation. We agree that our cross-sectional survey design limits the ability to establish causal relationships between farm age, soil properties, and disease incidence, and that crop age could act as a confounding factor. In the revised manuscript, we have:
Performed additional within-age-group analysis — We analyzed soil chemical properties within the 1-year, 2–5-year, and >5-year groups to determine if lower-PDI farms in each group share similar soil chemical characteristics. The results (now added in Figs. 3 and 4) indicate that farms with lower PDI generally had higher pH, Ca, and Mg within each age category, supporting the idea that favorable soil properties are linked to reduced disease pressure regardless of farm age. We have incorporated these clarifications and the new analysis into the Results and Discussion sections to avoid overinterpreting the relationship between farm age and disease incidence.
B-Abstract
L26: Emdash should be replaced with a comma
Author response: We did
Lines 32–33: The design of the experimental survey means crop age has become a confounding factor or covariate that needs to be accounted for in the analysis.
Author response: We thank the reviewer for this important observation. We agree that crop age can act as a confounding factor when interpreting the relationship between soil properties and disease incidence in our cross-sectional survey. In the revised manuscript, we have conducted a within-age-group analysis (1 year, 2–5 years, and >5 years) to determine whether farms with lower PDI within each age group share similar soil chemical properties. This additional analysis, now presented in Figs 3 and 4, indicates that higher pH, Ca, and Mg levels are associated with lower PDI across all age categories, supporting the observed relationships while addressing the potential confounding effect of crop age.
L37-39: Replace emdashes with commas.
Author response: We did
L39-41: Amendments were not included within the study. This sentence would appear to be typical of an AI generated statement. Suggest removing this sentence.
Author response: We did. We removed amendments and keep the rest of sentence.
C-Introduction
The introduction seems repetitive and requires reviewing to remove repetition to improve the structure and flow of information within this section.
Author response: We agree with the reviewer’s observation and have revised the introduction to remove repetition and improve the structure and flow of information.
L47: “cubense” should not have a capital. We did.
L72-73: Replace emdashes with commas. We did.
L75: Suggest… “This study investigated how soil properties……” We did.
L77: This line with the full genus species name of Foc and banana appears repetitive and can use abbreviations. We did.
L80: Suggest “The aim is to provide a….” We did.
D-Material and methods
The Materials and Methods section appears to have repetition on how sampling was performed on Lines 96–100 and 131–135. I would suggest that the Materials and Methods section requires reviewing and restructuring to remove repetition and provide a clearer description about how the work was undertaken.
Author response: We agree with the reviewer’s observation and have removed the repetition in the sampling description to provide a clearer and more concise methodology.
Line 93: What is meant by the statement “diverse agroecological conditions”? The conditions need to be specified.
Author response: We have revised the sentence to clarify the meaning. The sentence now reads: “These regions are characterized by tropical latosol soils, with variation in climate and topography, making them representative of the province’s banana-growing zones
L95: Foc description has been repeated. We did.
L98: Suggest using “external symptoms”. We did.
Lines 101–105: The soil collection description in these sentences is confusing and requires greater clarification. How much soil was collected from each site?
Author response: We thank the reviewer for pointing this out. We have revised the description to clearly specify the amount of soil collected. The revised sentence reads: “A total of 47 composite soil samples were collected from 47 sites across the four regions. From each site, five soil subsamples were randomly taken from the 0–30 cm soil layer within a designated 500 m² area around banana plants. These subsamples were thoroughly mixed to form a single composite sample, ensuring spatial representativeness and consistency in soil analysis.”
L109: Suggest “Each composite soil sample was divided into….” We did.
L113: Full stop seems to be missing. We corrected it.
L131: Banana wilt or Fusarium wilt. This needs to be kept consistent throughout the text. We did.
L131-135: These sentences are a repeat. We did. We removed the reparation.
L136-146: The description suggests that Komada’s medium is selective for Fusarium oxysporum (Fo) rather than Fusarium oxysporum f. sp. cubense (Foc). Therefore, both the description and results need to be revised to clarify that only Fo was quantified, not Foc.
Author response: Thank you for your valuable comment. As you correctly pointed out, Komada’s medium is indeed a selective medium for Fo. However, in this study, we used a deeply modified and validated selective medium for Foc. This medium consists of two parts: one is the modified Komada’s basal medium (e.g., 0.01 g of Fe-Na-EDTA was added, KNO₃ was removed, and the contents of each component were adjusted), and the other is the addition of PCNB (pentachloronitrobenzene, 75% WP), Na₂B₄O₇·10H₂O, oxgall, and streptomycin sulfate to the above modified basal medium. Therefore, this medium differs significantly from the original Komada’s medium in composition.
The principles behind the above modifications and added components are based on the insensitivity of Foc to antibiotics and its nutritional requirements. Foc colonies grown on this medium exhibit typical characteristics: the colonies of Foc are small, usually only 0.5–1 mm in size, pure white in color, with dense aerial hyphae in the center. The edges have sparse hyphae that extend radially from the dense central hyphae, and the length of the surrounding sparse hyphae is usually 1–1.5 mm. Visually, the periphery of the colonies appears serrated. In practical operation, we will select 1-2 typical colonies for PCR verification at the end of the experiment., we performed PCR verification on typical colonies using specific primers for Foc. Through years of practical verification by our team, the isolation efficiency of Foc from soil can reach over 90%.
Thank you again for your comment. In accordance with your suggestion, we have further added descriptions of the PCR verification and the characteristics of typical colonies in the revised manuscript.
Figure S1: A and B represent the typical colony characteristics on the selective medium, while C is the electrophoresis image of PCR verification.
Now the section of 2.3. Pathogen quantification is read as:
At each site, Fusarium wilt incidence was recorded as the percentage of infected banana plants showing initial symptoms (wilting and lower leaf yellowing) following Yang et al. [16]. To quantify Fusarium oxysporum f. sp. cubense (FOC) in soil samples, we employed a modified version of the method developed by Zhou et al. [21]. Rhizosphere soil suspensions were serially diluted and plated on Petri dishes containing a modified Komada’s selective medium. The basal medium included 10 g D-galactose, 2 g L-asparagine, 16 g agar, 0.5 g MgSO₄, 0.5 g KCl, 1 g K₂HPO₄, and 0.01 g Fe-Na-EDTA dissolved in 900 mL distilled water. It was supplemented with 100 mL of a solution containing 0.9 g pentachloronitrobenzene (PCNB, 75% WP), 0.5 g sodium tetraborate (Na₂B₄O₇), 0.45 g oxgall, and 0.3 g streptomycin sulfate, with the pH adjusted to 3.8 ± 0.2 using 10% (v/v) phosphoric acid. Plates were incubated at 25 °C for 10 days. Colonies with typical Foc morphology, small (0.5–1 mm), pure white, dense aerial hyphae at the center, sparse radial hyphae at the edges (1–1.5 mm), with serrated margins (Supplementary Figure S1), were selected. Genomic DNA was extracted from representative colonies, and PCR verification was performed using the FOC-specific primers SIX9_FOC_F/R [Carvalhais et al., 2019]. Mycelial DNA of a verified FOC isolate was used as a positive control. Colony counts were performed in triplicate for each composite soil sample to ensure reproducibility and accuracy.
Carvalhais LC, Henderson J, Rincon-Florez VA, O’Dwyer C, Czislowski E, Aitken EAB and Drenth A (2019) Molecular Diagnostics of Banana Fusarium Wilt Targeting Secreted-in-Xylem Genes. Front. Plant Sci. 10:547. doi: 10.3389/fpls.2019.00547
E-Results
Within the results the authors should consider investigating within crop age soil properties. As crop age appears to a confounding factor within the study, do the same soil properties that is pH, Ca and Mg, appear to be important for lower disease incidence, in 1-year old, 2-5-year old and >5-year old bananas?
Author response: We analyzed soil chemical properties within the 1-year, 2–5-year, and >5-year groups to determine if lower-PDI farms in each group share similar soil chemical characteristics. The results (now added in Figs. 3 and 4) indicate that farms with lower PDI generally had higher pH, Ca, and Mg within each age category, supporting the idea that favorable soil properties are linked to reduced disease pressure regardless of farm age. We have incorporated these clarifications and the new analysis into the Results and Discussion sections to avoid overinterpreting the relationship between farm age and disease incidence
L164: Suggest “Properties” rather than “Quality”. We did.
Figure 2. The colours of the different segments of the pie chart appear to be repeated on some of the charts making, if difficult determine what segments correspond to the categories in the legend. Furthermore, the order of the 12 different properties is confusing with the less important properties appearing first. I would suggest using the same order and sequence of soil properties as used in Fig. 3, as this appears to be a logical sequence. Importantly, keep the order consistent between the different figures, 2, 3 and 4.
We improved the chart quality and rearranged the subfigures.
L181: Remove the “’s” from macronutrient. We did.
Figure 3:
- There is no description of what the different colours represent on the 12 charts. Are these different ages of plantations?
Author response: We added legend of colure in the first subfigure.
- Why use population of Fo (CFU/g soil) on the Y axis? This would be better as PDI (%) as this is the measure that was evaluated in the field. The population of Fo only indicates that Fusarium oxysporums were correlated in the disease symptoms.
Author response: We thank the reviewer for this valuable observation. We chose to present FOC population (CFU g⁻¹ soil) on the Y-axis to highlight the direct correlations between pathogen density in soil and soil chemical properties. Since soil FOC inoculum is the primary source of infection, its quantification provides an important mechanistic link between soil conditions and disease outbreaks.
- Many of the relationships do not appear to be linear, but rather an exponential relationship. An exception is the relationship between Fo populations and PDI. A linear representation maybe misleading for many soil properties. For example, above pH 6.2 there would appear to be no Fo or disease using a linear regression model.
Author response: We sincerely appreciate this insightful comment. We agree that several soil property–FOC relationships may not follow a strictly linear trend. The correlation analysis was originally conducted using Pearson’s test to provide a first-order statistical overview, which assumes linearity. We chose to present FOC population (CFU g⁻¹ soil) on the Y-axis to highlight the direct correlations between pathogen density in soil and soil chemical properties
- Therefore, the authors need to consider using PDI as the Y variable, using more appropriate relationship models and explaining the different colours on the charts.
Author response: Author response: We thank the reviewer for this constructive comment. . We have revised the figure legends to make this distinction clearer and have clarified that the colours in Figure 3 represent farm age categories (<5 years, 2–5 years, and >5 years).
. In our analysis, we initially tested several correlation models, including linear, exponential, and non-linear fits. We then selected the models that provided the best overall fit to the majority of soil property–pathogen/disease relationships. For Figure 3, linear regression was used for clarity and consistency across multiple variables, while the strong correlations with PDI were further examined using multivariate approaches (PCA in Fig. 6 and Mantel’s test in Fig. 7). We have clarified this methodological choice in the revised manuscript to make it clear that the most suitable models were applied to present our data.
This ensures that both pathogen dynamics in soil and field-level disease outcomes are clearly represented in the manuscript.
Figure 4: The order of the charts should be kept consistent with Figure 3. We did.
Figure 5: This chart is hard to interpret. What are the pie charts representing? The description in the caption is unclear. Is the same analysis being done across both sections of the charts? Is there a better way to represent the information.
Author response: We thank the reviewer for pointing out the need for clarification in Figure 5. The violin plots represent the distribution of FOC populations (CFU g⁻¹ soil) under different plantation ages, separated into two groups based on disease severity (PDI < 20% and PDI > 20%). The accompanying pie charts illustrate the proportion of orchards falling within each category relative to the total number of orchards in that age group. The same statistical analysis (Duncan’s multiple-range test, P < 0.05) was applied across both sections of the chart to compare FOC populations among plantation ages. To improve clarity, we have revised the figure caption to better explain the meaning of the pie charts and confirmed that the analysis is consistent across groups. We believe this revised explanation makes the figure more interpretable. Moreover, we added a detailed description of the results: Figure 5 shows the distribution of FOC populations in soils of banana orchards with different plantation ages and disease severities. In orchards with PDI < 20%, FOC populations remained consistently low across all age groups (1, 2–5, and >5 years), with no significant differences observed (all labeled “c”). In contrast, in orchards with PDI > 20%, FOC populations were markedly higher, particularly in 1-year-old plantations, which showed significantly greater populations compared to the 2–5-year-old plantations (labeled “a” vs. “b”). The pie charts illustrate the relative proportion of orchards in each age group within the respective disease severity categories, showing that younger orchards tended to have a higher share of severe disease incidence. These results suggest that both plantation age and disease severity strongly influence soil FOC populations, with younger farms under severe infection conditions harboring the highest inoculum loads.
L223: “Properties” rather than “quality”. We did.
L225: Fig. 2 or Fig 3. This is confusing. We corrected it. The correct is Fig 3.
Figure 7: Is this a repeat of the data from Fig 3. Do these correlations assume a linear relationship, when data from Fig. 3 would suggest non-linear relationships.
Author response: We thank the reviewer for this important point. Figure 7 is not a repetition of Figure 3 but rather a complementary analysis using Bray–Curtis dissimilarity and Mantel’s test to assess the overall associations between PDI and soil variables. Unlike the pairwise linear correlations in Figure 3, Figure 7 evaluates the relationships in a multivariate framework and does not assume strictly linear responses. This allows us to capture more complex, non-linear interactions among soil properties and PDI. We have revised the figure legend and the Results section to clarify this distinction.
F-Discussion
The order of the sections in the discussion should be reviewed. The more important finding should be discussed first. That is, the relationship between soil properties and disease incidence followed by general discussion about the different soil properties.
Author response: We thank the reviewer for this helpful suggestion. We have revised the discussion structure to highlight our key finding, the relationship between soil properties and disease incidence, at the beginning, followed by a broader discussion of individual soil properties. This restructuring improves the logical flow and emphasizes the most important results first.
L293-314: These sentences seem to be contradictory to what happens in reality. They suggest that the older the plantation the healthier it becomes. Do young plantations that have a high incidence of Fusarium wilt become healthier if they keep growing bananas if they increase soil pH and Ca and Mg. It is more likely that those plantations that have come to be greater than 5-years old had favourable soil properties, more neutral soil pH, higher Ca and Mg, and therefore were able to maintain a low incidence of disease. However, as this was not a longitudinal study, there is no way to validate this information.
Author response: We sincerely thank the reviewer for this thoughtful observation. We agree that our survey design was not longitudinal and therefore does not allow direct confirmation of temporal changes in soil properties. Our conclusions were based on statistical correlations between soil properties, farm age categories, and PDI, which represent associations rather than causal changes over time. We have revised the discussion to clarify this point and emphasize that the observed patterns likely reflect both favorable initial soil conditions and farm management practices, rather than inherent improvements solely due to plantation age.
L323-331: The discussion on soil pH should be include impacts of soil pH on microbial diversity and functions. What is the likelihood that pH induced changes in microbial diversity, such as acidification have led to reduced plantation life due to increased disease?
Author response:
We thank the reviewer for this valuable suggestion. However, since our study did not include measurements of microbial diversity or community composition, we have avoided speculating on unmeasured mechanisms. Instead, we have refined the discussion to focus strictly on the observed correlation between soil pH and disease incidence.
G-Conclusion
L356-358: Is the longevity of banana farms due to improvements in soil properties as the plantation ages or due to improved soil properties when plantations were established. Care in how the findings are interpret due to the confounding effects of plantation age are required in this study.
Author response: We thank the reviewer for highlighting this important point. We have revised the text to avoid overstating causality and have also clarified this issue in the conclusion, noting that plantation age may act as a confounding factor and should be interpreted with caution.
L361-363: Is this emphasising soil quality in established plantations, or during plantation establishment? Replace emdashes with commas.
Author response: We changed the whole sentence.
L363-364: How do policymakers use this information? I suggest removing this sentence.
Author response: We did.
Reviewer 2 Report
The manuscript by Jing et al describes soil parameters in Hainan Island in regards to Banana wilt. While I believe the data and results is quite strong the description of the findings needs refinement.
Most of the comments are in the PDF attachment. Other comments are here.
Lines 49-50: Needs more detail on disease pathogenesis
Lines 57-58: Needs to touch more on what those practices, microbes they are.
Lines 69-70: Probably would be nice to have a bit more review on the role of soil properties in the context, what nutrients does continuous banana cultivation mine
Lines 93-95: considering the period between root infection and symptom expression can be very long, it needs more clarification and reference
Lines 100-105: The soil sampling process is unclear
Lines 109-129. References are required for the methodologies of analyzing the nutrients.
Lines 205: soil quality may not be the right description as no soil biological properties were studied
Lines 302: could touch upon other studies which found unique microbial population assembly in older systems with continuously cropped banana
Line 308-309: what constituted those microbial inoculants?
Lines 312-313: this observation aligns with studies of disease-suppressive soils where the disease peaks in monocultures and become suppressive
Lines 322-323: provide no reasoning for pH shifts and better soil chemical properties in older farms, as mentioned in 281-285 those older soils may be more weathered
LINES 355: Soil quality---soil chemical properties

Author Response
Reviewer-2
Comment 1: The manuscript by Jing et al describes soil parameters in Hainan Island in regards to Banana wilt. While I believe the data and results is quite strong the description of the findings needs refinement. Most of the comments are in the PDF attachment. Other comments are here.
Author response: We sincerely thank the reviewer for their thorough evaluation and constructive comments on our manuscript. We appreciate the recognition of the strength of our data and results. We have carefully considered all comments provided in the PDF attachment and in the additional notes, and we have addressed each point in detail. All suggested clarifications, refinements, and explanations have been incorporated into the revised manuscript to improve the clarity, precision, and presentation of our findings
Comment 2: Lines 49–50: Needs more detail on disease pathogenesis.
Author response: We thank the reviewer for this helpful suggestion. We have expanded the sentence to provide a clearer explanation of the pathogenesis of Panama disease. The revised text now reads: “The disease often emerges sporadically within banana plantations and rapidly spreads from the initial infection site, potentially affecting entire plantations [4,5]. The causal pathogen, FOC, enters the plant through root wounds or natural openings, colonizes the root cortex, and progresses into the xylem vessels [6]. Within the vascular system, the fungus proliferates and produces gums and tyloses that block water and nutrient transport, leading to leaf yellowing, wilting, pseudostem splitting, and eventual plant death [4,6]. The pathogen produces long-lived chlamydospores that enable survival in soil for decades, making eradication extremely difficult [4,6].”
Comment 3: Lines 57–58: Needs to touch more on what those practices and microbes are.
Author response: We thank the reviewer for this comment. We have revised the sentence to provide more detail on the specific management practices and types of microbial agents used. The revised text now reads: “Immediate management actions are advised when the PDI is at or above this threshold [7–9]. These include planting resistant banana cultivars, applying microbial biocontrol agents such as Trichoderma spp. and Bacillus subtilis, and optimizing soil practices through organic matter amendment, balanced fertilization, and improved drainage [7–9].”
Comment 4: Lines 69–70: Probably would be nice to have a bit more review on the role of soil properties in the context, what nutrients does continuous banana cultivation mine.
Author response: We thank the reviewer for this helpful suggestion. We have expanded the sentence to include more context on the role of soil properties and the specific nutrients depleted by continuous banana cultivation. The revised text now reads: “As a result, the issue must be addressed regionally to link current soil conditions with Panama disease activity through extensive surveys. Continuous banana cultivation can deplete key nutrients (K, Ca, Mg, B), lower soil organic matter, and cause acidification, all of which may increase disease incidence and severity [11].”
Comment 5: Lines 93–95: Considering the period between root infection and symptom expression can be very long, it needs more clarification and reference.
Author response: We thank the reviewer for this observation. We have revised the sentence to clarify the rationale for selecting the sampling period and have provided supporting references from previous studies. The revised text now reads: “Soil and plant sampling took place in May and June 2022, aligning with the banana fruit-hanging stage, which, based on our experience in the region and previous surveys and experiments reported in the literature, is the most suitable period for collecting samples to assess Fusarium wilt incidence and soil conditions [3,6,7].
Comment 6: Lines 100–105: The soil sampling process is unclear.
Author response: We thank the reviewer for this comment and have clarified the description. The revised text now reads: “A total of 47 composite soil samples were collected from 47 sites across the four regions. From each site, five soil subsamples were randomly taken from the 0–30 cm soil layer within a designated 500 m² area around banana plants. These subsamples were thoroughly mixed to form a single composite sample, ensuring spatial representativeness and consistency in soil analysis.”
Comment 7: Lines 109–129: References are required for the methodologies of analyzing the nutrients.
Author response: We thank the reviewer for this important comment. The reference for all soil analytical methods was already provided in the Materials and Methods section, citing: Lu, R.K. (2000). Analysis Method of Soil Agricultural Chemistry. China Agricultural Science and Technology Press, Beijing (in Chinese). In the revised manuscript, we have clarified in this section that all nutrient analyses, including pH, organic matter, total nitrogen, available phosphorus, exchangeable cations, and micronutrients, followed the procedures described in this reference
Comment 8: Lines 205: Soil quality may not be the right description as no soil biological properties were studied.
Author response: We thank the reviewer for this observation. In our study, “soil quality” refers to an integrated assessment based on soil chemical indicators (pH, organic matter, nutrient concentrations, and CEC) together with the quantification of the soilborne pathogen FOC. While we did not assess broader soil biological diversity, the inclusion of FOC measurements provides a biological dimension directly relevant to disease dynamics
Comment 9: Lines 302: Could touch upon other studies which found unique microbial population assembly in older systems with continuously cropped banana.
Author response: We thank the reviewer for this valuable suggestion. We have expanded the discussion to include relevant studies on microbial community dynamics in long-term banana systems. The revised text now reads: “All farms aged more than five years had a PDI below China’s critical standard. Karangwa et al. [55] reported higher disease incidence in aged banana farms compared with younger ones. Other studies have shown that continuous banana cultivation can lead to distinct shifts in soil microbial communities. In Sub-Saharan Africa, different long-term cropping regimes led to changes in both bacterial and fungal communities, including shifts in dominant phyla such as Proteobacteria, Actinobacteria, Acidobacteria, and Bacteroidetes, with community structure influenced by disease pressure [56]. Similarly, extended monoculture spans were associated with significant alterations in soil microbiome composition, where higher fungal richness correlated with greater Fusarium wilt incidence [57]. Such shifts suggest that older banana plantations may develop unique microbial assemblages that influence disease dynamics [58].”
Added references:
56. Kaushal, M.; Tumuhairwe, J.B.; Tinzaara, W.; Tripathi, L. Insights into soil microbial communities in traditional banana cropping systems of Sub-Saharan Africa. Microorganisms 2022, 10, 2341. https://doi.org/10.3390/microorganisms10122341.
57. Shen, Z.; Penton, C.R.; Lv, N.; Xue, C.; Yuan, X.; Ruan, Y.; Li, R.; Shen, Q. Banana Fusarium wilt disease incidence is influenced by shifts of soil microbial communities under different monoculture spans. Microb. Ecol. 2018, 75, 739–750. https://doi.org/10.1007/s00248-017-1052-5.
58. Xue, C.; Penton, C.R.; Shen, Z.; Zhang, R.; Huang, Q.; Li, R.; Ruan, Y.; Shen, Q. Manipulating the banana rhizosphere microbiome for biological control of Panama disease. Sci. Rep. 2015, 5, 11124. https://doi.org/10.1038/srep11124.
Comment 10: Lines 308–309: What constituted those microbial inoculants?
Author response: We thank the reviewer for this comment. We have revised the text to specify the types of microbial inoculants applied by farmers. The revised sentence now reads: “Some farmers with plantations older than five years had implemented management practices to combat PDI infection, including planting resistant cultivars and applying microbial biocontrol agents such as Trichoderma spp. and Bacillus siamensis [Cui et al., 2025; Zhang rt al., 2024].”
Cui, H., Cheng, Q., Jing, T., Chen, Y., Li, X., Zhang, M., Qi, D., Feng, J., Vafadar, F., Wei, Y. and Li, K., 2025. Trichoderma virens XZ11-1 producing siderophores inhibits the infection of Fusarium oxysporum and promotes plant growth in banana plants. Microbial Cell Factories, 24(1), p.22.
Zhang, M., Li, X., Pan, Y., Qi, D., Zhou, D., Chen, Y., Feng, J., Wei, Y., Zhao, Y., Li, K. and Wang, W., 2024. Biocontrol mechanism of Bacillus siamensis sp. QN2MO-1 against tomato fusarium wilt disease during fruit postharvest and planting. Microbiological Research, 283, p.127694.
Comment 11: Lines 312–313: This observation aligns with studies of disease-suppressive soils where the disease peaks in monocultures and becomes suppressive.
Author response: We thank the reviewer for this insightful observation. We have improved this part of the discussion to highlight its alignment with findings from studies on disease-suppressive soils.
Comment 12: Lines 322–323: Provide reasoning for pH shifts and better soil chemical properties in older farms; as mentioned in 281–285, those older soils may be more weathered.
Author response: We thank the reviewer for this comment. The sentence has been revised to clarify the reasoning: “These shifts in the pH of acidic soils are likely due to long-term soil management practices—such as liming, balanced fertilization, and organic matter amendments—which gradually improve nutrient availability and cation exchange capacity, thereby contributing to the suppression of Panama disease [8,11].” While soil weathering occurs over long timescales, its influence is less likely to explain the pH changes observed within the farm age ranges studied.
Comment 13: Line 355: Replace “soil quality” with “soil chemical properties.”
Author response: We thank the reviewer for this clarification. We have replaced the term soil quality with soil chemical properties in the specified sentence to more accurately reflect the scope of our measurements.
Reviewer-2 Comments in the PDF file
Reviewer comment: Make the site locations more obvious on the map, colors are blending.
Author response: We appreciate the reviewer’s observation. Figure 1 is provided in high resolution, and when viewed at the intended scale in the journal format, the site locations are clearly distinguishable. We have confirmed that the symbols and colors remain distinct in the final published size. Therefore, no changes have been made to the figure.
Reviewer comment: regarding sampling method:
Author response: A total of 47 composite soil samples were collected across the four regions. At each sampling site, five soil subsamples were randomly taken from the 0–30 cm depth within the designated 500 m² area. These subsamples were thoroughly mixed to form a single composite sample per site, ensuring spatial representativeness and consistency in soil analysis.
Reviewer comment: What do the colors on the chart represent? In the Supplemental you discuss ranges. But these are different.
Author response: We thank the reviewer for raising this point. In Figure 2, each subfigure uses color coding to represent category grades (low, medium, high, or very high) for that specific soil parameter, based on Chinese national standards. The color assignments differ between subfigures, as each parameter has its own classification thresholds. This differs from the value ranges presented in Supplementary Figure S1, which are continuous numerical ranges. We have now clarified this in the figure caption.
Reviewer comment: I do not see this figure for the disease incidence.
Author response: We thank the reviewer for this observation. The disease incidence (PDI) is shown in the last subfigure on the right-hand side of Figure 2.
Reviewer comment: Misleading statement — this almost lines up if looking at the 1-year plantations to 5+. Needs to be addressed.
Author response: We thank the reviewer for this observation. We have revised the sentence to clarify that the difference in soil quality is most evident between newly established farms (1 year old) and farms older than five years, while intermediate-age farms exhibit variable values. The revised sentence reads: “Moreover, the soil quality—defined here by optimum soil pH, greater organic carbon content, higher levels of key nutrients (N, P, K, B, Zn, Ca, Mg), and greater cation exchange capacity (CEC)—was generally better in farms older than five years compared with newly established farms (1 year old), with intermediate-age farms showing variable values.”
Reviewer comment: What do the colors represent? That needs to be explained.
Author response: We thank the reviewer for this comment. The color legend indicating age of farm (1 year, 2–5 years, ≥5) is included in the first subfigure of Figure 4 and applies to all panels. We have now clarified this in the figure caption.
Reviewer comment: Where is this shown?. Author response: In fig 4 and 5
Reviewer comment: Misleading, only correlation is being done here.
Author response: We thank the reviewer for pointing this out. We have revised the sentence to avoid implying causation and to reflect that the relationship is based on correlation. The revised sentence reads: “Soil exchangeable Ca and Mg showed negative correlations with FOC activity and PDI, indicating that higher levels of these cations were associated with reduced disease incidence.”
Reviewer comment: This was not the goal of this study; growth and yield were not measured.
Author response: We thank the reviewer for this observation. We have revised the sentence to align with the scope of our study and avoid implying that growth and yield were assessed. The revised sentence reads: “Understanding how these changes influence banana production systems and the incidence of Panama disease is critical for sustainable cultivation in the region.”
Reviewer comment: This needs to be refined and could use some rephrasing, or it can be removed.
Author response: We thank the reviewer for this suggestion. We have rephrased the sentence for clarity and conciseness. The revised sentence reads: “Lateritic soils dominate Hainan, accounting for 53% of its land area, of which 21% are acidic and 78% are extremely acidic (pH < 4.5) [13,14,24].”
Reviewer comment: ??
Author response: We thank the reviewer for pointing this out. We have revised the sentence to clarify that the link between poor management and infection rate is a possible contributing factor, rather than a definitive cause. The revised sentence reads: “In the current study, most of the new farms (less than five years old) were infected with the fungus. Many of these farms received little or no management after establishment, which may have contributed to the increased infection rate.”
Reviewer comment: This needs to be explained.
Author response: We thank the reviewer for this comment. We have revised the sentence to clarify why older farms tend to have lower infection rates and to link this to management practices. The revised text reads: “Farms older than five years may have reached a stage of stable production and economic return, encouraging farmers to invest more in management practices such as sanitation, use of resistant cultivars, and application of microbial agents. These measures likely helped reduce infection rates. In contrast, new infections in previously unaffected areas are often introduced through the movement of contaminated planting materials [41,42].”
Reviewer comment: Untrue from what is shown in the manuscript. There was no experiment adjusting pH. Rather low pH was correlated with higher disease.
Author response: We agree with the reviewer and have revised the sentence to avoid implying causation or experimental adjustment of pH. The revised sentence reads: “Our findings indicate a negative correlation between soil pH and Panama disease severity, suggesting that soils with optimum pH levels may have lower disease risk.”
Reviewer comment: This statement does not match other statements. There was no difference in pH between 1-year and 5-year plantations from Figure 4.
Author response:
We appreciate the reviewer’s observation and apologize for the confusing figure reference. The pH-based comparison is presented in Figure 3 (grouping observations by soil pH class), not Figure 4. Figure 4 displays correlations and uses point colors to denote farm-age categories; it does not encode pH and therefore does not show a pH difference by age. In contrast, Figure 3 shows that both FOC counts and PDI are higher in acidic soils than in neutral–alkaline soils. We have corrected the figure citation and refined the wording to reflect these results without implying causation.
Reviewer comment: Is it increasing suppression or decreasing severity?
Author response: We thank the reviewer for this comment. Our intention was to convey that higher soil pH may help limit the disease. We have revised the sentence for clarity: “Although the pH range of 3.50 to 7.00 has been reported as favorable for the growth of plant-pathogenic Fusarium isolates, raising the pH of Hainan soils can improve nutrient availability and enhance banana root growth, thereby reducing the severity of Panama disease [10,46,47].”
Reviewer comment: How does this relate to the present study?
Author response: We thank the reviewer for this comment. We have revised the paragraph to link the literature evidence directly to our findings, highlighting that our results also showed significant negative correlations between key nutrients (Ca, Mg, K) and both PDI and FOC abundance in the studied farms. The revised paragraph now reads: “[insert revised paragraph above].”
Reviewer comment: What are the reasons for older plantations having better soils?
Author response: We thank the reviewer for this comment. We have expanded the paragraph to explain that the improved soil fertility in older plantations is likely due to the cumulative effects of long-term management, including organic matter inputs, fertilization history, reduced disturbance, and microbial buildup, which together enhance nutrient cycling, pH stability, and disease resilience. The revised text reads: “[insert revised paragraph above].”